# A quantitative modelling approach for DNA repair on a population scale

**Leo Zeitler**, **Cyril Denby Wilkes**, **Arach Goldar**, **Julie Soutourina** *

Université Paris-Saclay, CEA, CNRS, Institute for Integrative Biology of the Cell (I2BC), Gif-sur-Yvette, France

* julie.soutourina@cea.fr

**Data Availability Statement:** All code that was used for the study is available on https://github.com/leoTiez/jmak.

**Funding:** LZ was supported by the CEA NUMERICS program, which has received funding from

## Abstract

The great advances of sequencing technologies allow the *in vivo* measurement of nuclear processes—such as DNA repair after UV exposure—over entire cell populations. However, data sets usually contain only a few samples over several hours, missing possibly important information in between time points. We developed a data-driven approach to analyse CPD repair kinetics over time in *Saccharomyces cerevisiae*. In contrast to other studies that consider sequencing signals as an average behaviour, we understand them as the superposition of signals from independent cells. By motivating repair as a stochastic process, we derive a minimal model for which the parameters can be conveniently estimated. We correlate repair parameters to a variety of genomic features that are assumed to influence repair, including transcription rate and nucleosome density. The clearest link was found for the transcription unit length, which has been unreported for budding yeast to our knowledge. The framework hence allows a comprehensive analysis of nuclear processes on a population scale.

## Author summary

As DNA encodes our very identity, it has been subject to a plethora of studies over the last century. The advent of new technologies that permit rapid sequencing of large DNA and RNA samples opened doors to before unknown mechanisms and interactions on a genomic scale. This led to an in-depth analysis of several nuclear processes, including transcription of genes and lesion repair. However, the applied protocols do not allow a high temporal resolution. Quite the contrary, the experiments yield often only some few data signals over several hours. The details of the dynamics between time points are chiefly ignored, implicitly assuming that they straightforwardly transition from one to another. Here, we show that such an understanding can be flawed. We use the repair process of UV-induced DNA damage as an example to present a quantitative analysis framework that permits the representation of the entire temporal process. We subsequently describe how they can be linked to other heterogeneous data sets. Consequently, we evaluate a correlation to the whole kinetic process rather than to a single time point. Although the approach is exemplified using DNA repair, it can be readily applied to any other

European Union's Horizon 2020 research and innovation program under the Marie Sklodowska-Curie grant agreement No 800945. The work was supported by the Fondation ARC (PGA1 RF20170205342) and Comité Ile-de-France - La Ligue Nationale Contre le Cancer. The funders had no role in study design, data collection and analysis, decision to publish, or preparation of the manuscript.

**Competing interests:** The authors have declared that no competing interests exist.

mechanism and sequencing data that represent a transition between two states, such as *damaged* and *repaired*.

This is a *PLOS Computational Biology* Methods paper.

## Introduction

As DNA represents the hereditary unit of life, maintaining its integrity is vital for every organism's survival. A large variety of different genotoxic factors have the potential to damage the molecular structure of DNA. Among others, it has been shown that UV light induces Cyclobutane Pyrimidine Dimers (CPDs). Nucleotide Excision Repair (NER) is an evolutionarily conserved repair mechanism in *Saccharomyces cerevisiae* that can remove a broad range of damage, including CPDs [1]. NER is conventionally divided into two subpathways. The first recognition mechanism is named Global-Genome Repair (GGR) and can be observed along the entire genome. DNA damage is recognised directly by protein association. There is evidence that protein loading is promoted through interactions with chromatin remodellers that change the nucleosome density or distribution [2, 3]. The second pathway is restricted to actively transcribed regions; hence the name Transcription-Coupled Repair (TCR). Expressed genes exhibit quicker repair than silent downstream regions [4]. This promoted the assumption that TCR is more efficient than GGR, although constrained to the transcribed strand (TS) [5, 6]. TCR is initiated by lesion-blocked RNA polymerase II (Pol II) which cannot continue elongation [7]. Thus, a potential link of TCR to transcription rate has been indicated by several studies [8, 9]. After recognition, TCR and GGR use the same incision and nucleotide replacement mechanism. DNA is incised to either side of the lesion leaving an approximately 30-nucleotide gap, which is subsequently replaced and ligated (for a comprehensive description and analysis, see the review by [10]).

Our understanding of such processes in living cells has been largely enhanced by Next Generation Sequencing (NGS). It allows the identification of enriched loci of a selected property on a genome-wide scale. Among others, it has been applied to investigate the CPD repair mechanisms *in vivo* through analysing temporal changes of the damage distribution. [8] obtained high-resolution CPD-seq data that are often used as a benchmark reference (see for example [9]). Their analysis indicates that single nucleosomes and DNA-bound transcription factors have an impact on the CPD formation. Moreover, they point out that repair is seemingly influenced by the CPD position with respect to the nucleosomal dyad as well as the transcription rate of genes. Another major contribution has been done by [9]. Their protocol for eXcision Repair sequencing (XR-seq) revealed strong TCR at early time points which is followed by repair in non-transcribed regions. Furthermore, [11] and [12] utilised CPD data to compute repair rates in different areas, which indicated that the process is highly organised into genomic regions. By using GGR-deficient strains, they show that repair is changing globally when the subpathway is repressed. This is compared to the distribution of repair proteins and histone modifications.

Unfortunately, due to costs and constraints in the experimental protocol, NGS data sets contain barely more than a few time points over several hours. Consequently, previous studies could only derive limited conclusions, e.g. the absolute change at different loci. We argue that

such an analysis ignores valuable information about the transitional process from one time point to another. Furthermore, it should be emphasised that sequencing signals are commonly understood as representing an average cell. We advocate an interpretation where the data is explained as the product of many independent cells. Thus, the repair dynamics are driven by non-interfering stochastic processes. Without assuming any specific molecular mechanism, we hypothesise that they are composed of two independent random variables namely accessibility to the lesion governing repair times and Brownian motions of proteins through the nucleus to find their target. It has been shown by several studies that proteins exhibit a range of different movements in the nucleus [13–18]. Diffusion has also been investigated and modelled in context of DNA repair for the Rad4-Rad23 complex [19]. Although protein movements have been used to understand specifics of NER kinetics, a framework to quantitatively describe population-based sequencing data is still lacking.

Our approach and main results can be summarised as follows. The sparse temporal resolution of NGS data sets makes it necessary to incorporate precise assumptions about the nature of the process in order to recover missing information. Here, we present a computational framework to analyse DNA repair kinetics. We derive a function to study CPD removal as a Poisson point process of independent cells. Since we do not impose any molecular mechanism, we obtain a simple and minimal representation. The parameters can be derived using the well-studied physical model for phase transitions, which is described in detail by Kolmogorov [20], Johnson and Mehl, [21], and Avrami [22–24] (KJMA model). It can be conveniently transformed to a linear regression problem and is therefore executable on almost any ordinary computer. A consequence that is implied by our repair model is that the observed change of CPDs is non-constant over time. To our understanding, this has not been explicitly incorporated in the analysis of NGS data. The model validity can be verified with independently probed XR-seq data [9]. We are able to recover particular aspects of the NER kinetics despite our broadly applicable assumptions. We ultimately use the framework to predict correlations with other nuclear processes. It is able to establish interrelationships that are supported by other studies such as nucleosome density [8, 12] and transcription rate [8]. It is most surprising, however, that we find the strongest correlation with transcription unit (TU) length, which is a new finding for budding yeast to our knowledge. Interestingly, our model allows also an alternative understanding of the data in which repair positions grow as patterns in a population. Although the analysis has been demonstrated for DNA repair, it can be applied to any process that can be modelled by an irreversible binary state transition. The source code is available on GitHub: https://github.com/leoTiez/jmak [25].

## Results

### Modelling DNA repair

In a single cell, CPD damage describes the mispairing of two adjacent pyrimidine nucleobases. Instead of establishing hydrogen bonds to the opposite strand, they cause two consecutive nucleotides to bind to each other. Consequently, there can be maximally one lesion per position. This results in a zero-one (i.e. *damaged-repaired*) state space per position and per cell. During ongoing repair, lesions are removed, and positions change subsequently their state to *repaired*. It can be assumed that this process is stochastic and involves to some extent unpredictable noise. If we could repeatedly and independently measure the repair times for a single position in a single cell, we could distribute the measurements over a timeline (Fig 1B). This type of data can be investigated by a Poisson point process, which allows the derivation of a predictive function that expresses the probability of repair over time. If we would temporally discretise the data over larger bins, all repair time points within a bin were aggregated together.

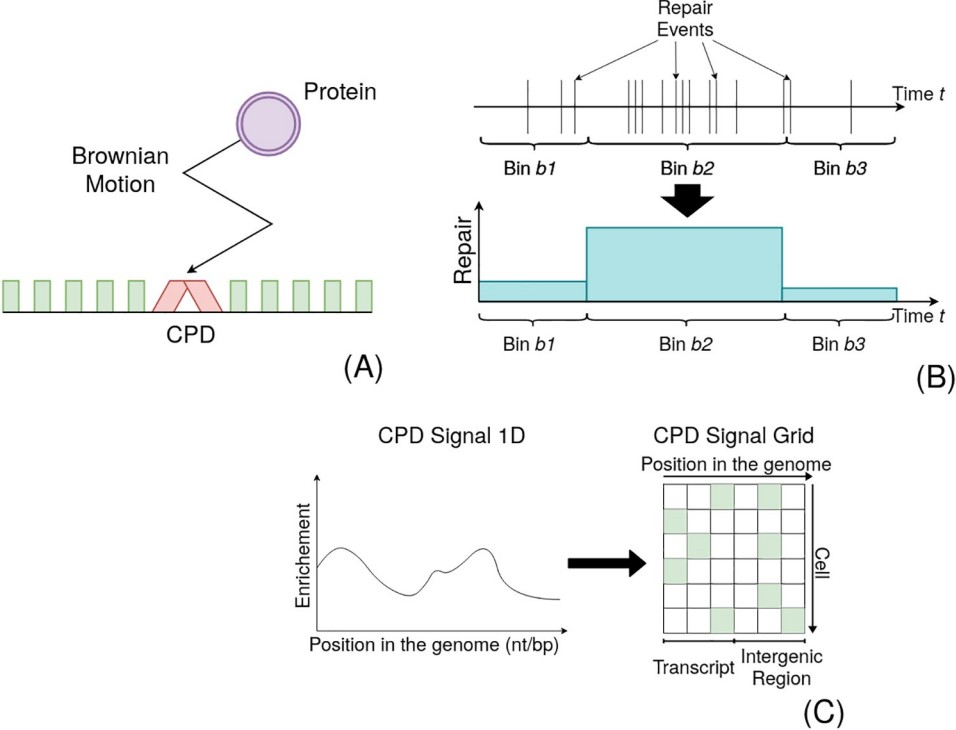

**Fig 1. Schematic representation of the Process.** (A) The repair proteins' search process (purple with arrow) to the lesion (red) can be understood as a Brownian motion. Repair happens after association with some delay, which we assume to be another random variable (waiting time). (B) If we could repeatedly measure the repair times for one position in a single cell individually, we could distribute the measurements over a timeline. This can be analysed by a stochastic point process. Binning the timeline should correspond to the observable change of CPDs in a given window. (C) We presume that this stochastic repair process happens independently in each cell. Therefore, CPD-seq data can be understood as an accumulation of several experiments. This allows an understanding of the signal as a two-dimensional grid. Since there can be only one lesion per position per cell, we understand the amplitude of the signal as a surrogate for the number of cells that contain a lesion at this locus (marked in green).

In the following, we assume that this is given by the change of the CPD-seq signal, as the amplitude decrease at an arbitrary position must explain the number of cells that have repaired their lesions. Consequently, the data represent the process over the entire cell population. We conjecture that the dynamics are independent between cells. The CPD-seq data can be therefore alternatively interpreted as a two-dimensional grid: one axis representing the cells and the other the nucleotide positions (Fig 1C). We understand NGS signals not as representing an average cell but the mutual effect of multiple independent cells.

It is clear that the number of cells with a lesion at a given position is discrete. Therefore, the number of repair events in a given time window can be studied with a Poisson process. It expresses the probability for a number of events $N(t) = n$ in a given time $t$ and takes the form $P(N(t) = n) = \frac{(\lambda t)^n}{n!} e^{-\lambda(t)t}$, where $\lambda(t)$ denotes the rate of events as a function of time. It is commonly assumed that CPD repair is irreversible if the irradiation source is removed. It follows that the probability that repair has happened ($p_r$) at time $t$ can be expressed by the cumulative distribution function for the Poisson process, namely

$$p_r = 1 - e^{-\int \lambda(t)dt}. \tag{1}$$

To find a functional representation of $\lambda(t)$, we conjecture that repair proteins move through random Brownian motions (diffusion) to the repair sites, subsequently associate to the DNA and remove the lesion. The entirety of this mechanism can be understood as a mixture of two random processes: diffusion and waiting/repair time (Fig 1A). We surmise that the waiting time is determined by the accessibility to the lesion. The mutual effect of repair proteins removing DNA lesions becomes observable through the decrease of the CPD-seq signal. We thus assume that this change $\Delta C$ during time $\Delta t$ is proportional to the searched volume by these proteins $D_{\hat{m}} t^{\hat{m}}$ (which is related to the mean squared displacement) and the average of the expected repair time $\hat{\beta}$

$$\lambda(\Delta t) = \Delta C \propto b \Delta t^{\hat{m}} + O(\Delta t^{\hat{m}}). \tag{2}$$

$b$ denotes a scaling factor that accounts for the diffusion constant $D_{\hat{m}}$ and $\hat{\beta}$. $t^{\hat{m}}$ with the the the anomalous coefficient $\hat{m}$ is the dominating term with the highest order. If $\hat{m} < 1$, the process is called subdiffusive; if $\hat{m} > 1$, the movement exhibits a superdiffusive behaviour. It is clear that the integral over Eq 2 also follows a power-law, i.e. $\int \lambda(t)dt \propto bt^m + O(t^m)$. Substituting in Eq 1 results in $p_r = 1 - e^{-bt^m}$. When setting $\sqrt[m]{b} = \beta = 1/\tau$ and assuming that only a fraction $\theta \in [0, 1]$ of cells have the ability to repair their lesions in a given time, we obtain

$$f(t) = \left(1 - \exp\left[-\left(\frac{t}{\tau}\right)^m\right]\right)\theta. \tag{3}$$

$\tau$ is the characteristic time until repair can be observed. An equation with a similar form to describe the phase transition in solids was derived independently by [20, 21], and [22–24] (KJMA model). As Eq 3 can be converted to a linear regression problem (Eq 5), it can be straightforwardly applied to find the necessary parameters. More interestingly, the KJMA model allows an alternative understanding of the data and the process, which is explained in Discussion.

## Applying the model to the data

We formulated three expectations in order to prove the model validity. Firstly, we required that the estimated repair dynamics need to be in line with independently probed data; secondly, we thought it to be indispensable to recover NER-specific features that were not implicitly incorporated into the model; and lastly, Eq 3 needs to make verifiable predictions about other factors that influence repair. CPD data for the parameter estimation were taken from [8] (0, 20, 60, and 120 minutes after irradiation) and divided into different segments where $\lambda(t)$ was assumed to be spatially constant. We distinguish between TCR regions, which are the TS of genes that presumably exhibit TCR; the NTS of TCR regions; and non-TCR areas, which are composed of transcripts where the effect of TCR is not evident and intergenic regions (see S1 Appendix as well as S1 and S2 Figs). Moreover, TS and NTS of TCR regions were equally divided into start, centre, and end. Subsequently, CPD data was converted to represent repair using Eq 7 (in the following also called repair data). An example of the predicted repair dynamics is given in S3 Fig. We also compared the presented results with the analysis of a more traditional segmentation into TS and NTS of all genes as well as intergenic regions (S3 Appendix).

XR sequencing provides a snapshot of currently ongoing repair in the cell culture, and it therefore represents an independent angle on CPD removal. It should correspond to the derivative of Eq 3 (given in Eq 6). XR-seq signals were taken from [9] (5, 20, and 60 minutes after irradiation) and segmented as for the CPD-seq data. We assumed that a surrogate for ongoing

repair can be additionally derived from the CPD-seq data themselves by calculating the damage decrease per time (S4 Appendix Eq 4). This was used as a baseline value for the correlation between model and data. As we assumed a non-linear interrelationship, we used the distance correlation (DC) as a correlation measurement (S4 Appendix). Strikingly, the predicted repair rates correlate clearly better (DC = 0.441) than the actual data (DC = 0.209) (compare Fig 2A with 2B; see S2 Table). Moreover, the model predictions align fairly well with the XR data (exemplified in S4 Fig). We hence surmise that Eq 3 is in agreement with independently probed data.

Despite the fact that we model time-dependent repair, we nevertheless do not incorporate two (potentially competing) repair mechanisms, i.e. TCR and GGR. It is commonly presumed that CPD removal through TCR is quicker than by GGR [4, 26]. Moreover, as GGR acts genome-wide, they are spatially non-exclusive for genes. Indeed, we can recover the cumulative effect when averaging the repair evolution for a group of segments, e.g. the start of TCR regions. The beginning of TCR areas is almost solely repaired by a single mechanism (Fig 3A). The contribution of this pathway is decreasing as a function of distance from the transcription start site (TSS). Instead, a later acting mechanism becomes increasingly observable (Fig 3B and 3C). Lastly, lesions in non-TCR regions are only detected by the late-acting process (Fig 3D). This is even the case despite the fact that non-TCR areas also include transcripts. We deduce that these two distinct pathways show the effect of TCR and GGR along the gene. We were therefore able to separate the effect of two distinct NER processes without involving any particular mechanism. Parameter distributions (i.e. $m$, $\tau$, and $\theta$) that create these repair kinetics are exemplified in Fig 4. Surprisingly, the NTS possesses different dynamics whilst not exhibiting any difference between start, centre, and end (S5 Fig). The average repair fraction is much lower than for all other areas ($\theta \approx 0.6$ instead of $\approx 0.8$). Moreover, we observe a subtle early increase of the derivative, indicating a larger presence of early repair in comparison to non-TCR regions. It is difficult to analyse this trend without additional experiments. It could simply be the impact of neighbouring overlapping regions. However, these results could equally point to different repair dynamics on the NTS.

Lastly, we extended the analysis to make predictions about influencing factors for CPD repair *in vivo*. Previous studies published various measurements of different nuclear properties that could possibly interact with lesion removal dynamics. To assess the predictive power of our model, we opted to analyse a link to transcription rate [8, 9] and nucleosome density as representing chromatin structure [8, 12, 27]. We also investigated a link to TU length and the relative distance to centromeres and telomeres as possible unreported affecting parameters. We used the NET-seq signal produced by [28] as a surrogate for transcription rate without UV irradiation. The TU length was measured by [29]. Nucleosome data after UV treatment were acquired by [12]. We excluded regions outside a reasonable parameter range from the subsequent analysis (see Methods and materials and S2 Appendix; the number of removed regions is given in S1 Table). Correlations with the model parameters were verified with a significance test, during which we compared a binary classification model with the performance of a random model (see Methods and materials and S5 Appendix for more information; the working of the classifier is explained in S6 Fig). Interrelationships to other sequencing data are elaborated and discussed in S6 Appendix. All data distributions are given in S7–S12 Figs. The results differed considerably depending on the genomic area and context (S3 Table). TCR has been repeatedly investigated with respect to transcription rate, and it is surmised to be positively correlated. Hence, higher transcription yields quicker repair, whereas low transcription results in slower lesion removal [8, 9]. Our model confirms these findings (Fig 4A and 4B). Non-TCR segments and the start of the TS seem to be starkly influenced by the nucleosome density, whereas all other areas do not show a strong correlation (Fig 4C and 4D). The clearest results,

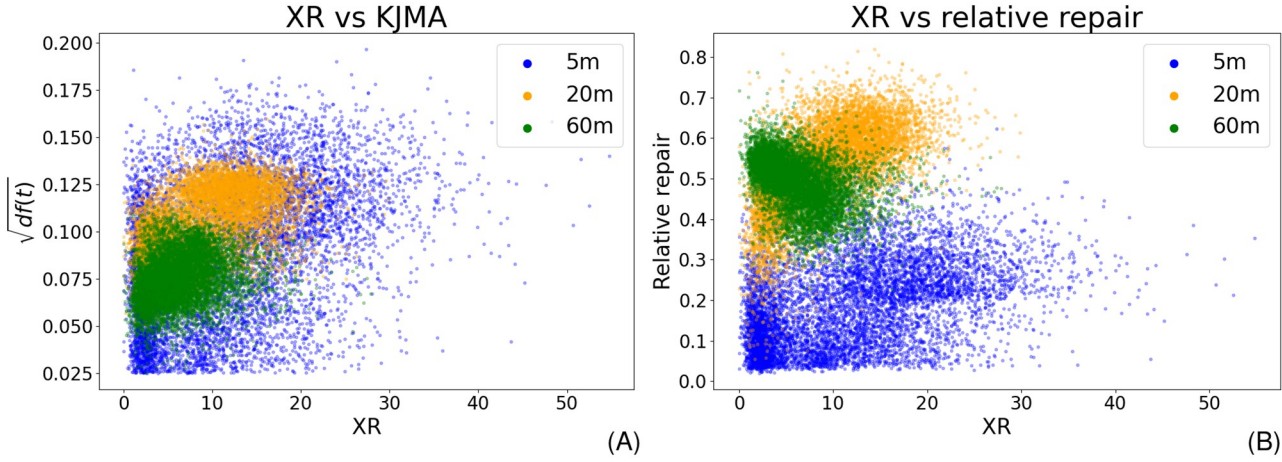

**Fig 2. Comparing XR-seq data with model predictions.** The values at 5 minutes are given in blue, 20 minutes are coloured yellow, and 60 minutes are green. The plots show that the distance correlation between prediction and XR-seq data is even higher than for the CPD-seq data. (A) Predicted repair rates with respect to the XR-seq data exhibit a considerably strong correlation (DC = 0.441). Predictions are given as the square root of the model prediction. This reduces the effect of increasing variance with larger derivatives. (B) The repair rates derive from the data as a function of XR-seq values shows a weaker correlation (DC = 0.209).

however, were obtained by the TU length. Both, TS (Fig 4E) and NTS (Fig 4F) are clearly influenced. The TU length is therefore likely to be contributing to the lesion removal dynamics. This is an unreported finding for budding yeast to our knowledge. The developed quantitative framework has hence the potential to identify established as well as new interrelationships. Importantly, a correlation with the distance to telomeres and centromeres (Eq 9) did not indicate a significant link. This shows that the applied method is selective for certain correlations (S12 Fig).

## Discussion

The few time points of NGS data sets require a temporal model to recover missing information between data samples. In this work, we developed a computational approach to describe the DNA repair kinetics on a population scale. We recover region-specific properties based on the genome-wide distribution of DNA damages. We assume a mixture of two stochastic processes (diffusion and lesion accessibility) that collectively explain the change of CPD data over time. Parameters of the derived equation can be estimated with the KJMA model, that is conveniently converted to a linear regression problem. This allowed the analysis of the temporal process as a whole rather than only comparing single time points. Importantly, it points out that the signal changes non-linearly over time. This is expected from a biological point of view, as TCR and GGR are commonly seen as acting within different time scales. However, it should be emphasised that it has not been incorporated in the analysis of temporal changes in sequencing data to our knowledge. The model therefore accounts specifically for dynamics on a population scale. Moreover, the derivative (Eq 6) provides key information about active ongoing repair. It thus permits linking CPD-seq data—showing the DNA damage distribution over the genome—and XR-seq data of excised DNA fragments generated by repair. This provides strong support for the validity of the model. Even though Eq 3 represents repair only with one mechanism per region, the combined effect of TCR shortly after UV treatment and GGR at later time points can be recovered when considering the average over several areas. The model can be readily used to uncover interrelationships between repair parameters and

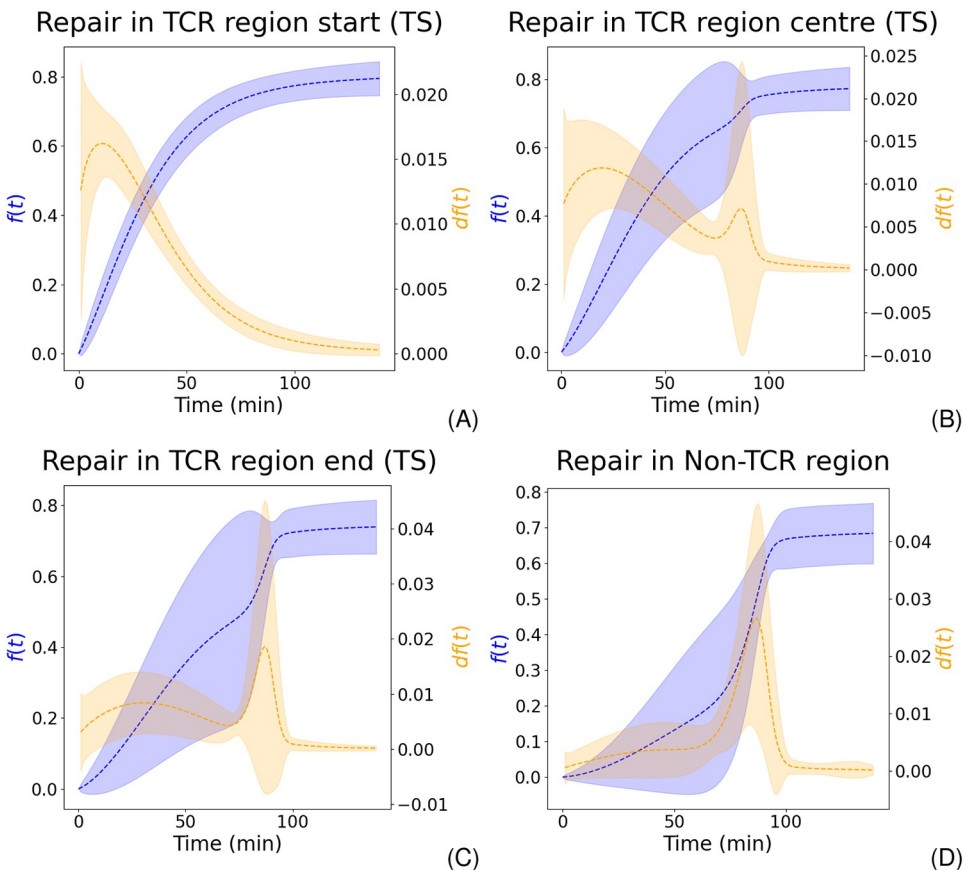

**Fig 3. Collective behaviour of genomic regions can recover mutual effect of TCR and GGR.** Dashed lines give the mean whereas the shaded areas show the standard deviation. Blue and orange represent the repair fraction and the repair rate (derivative of the repair fraction), respectively. (A) The start of TCR regions is repaired early after irradiation, demonstrating the effect of TCR. (B) At the centre of TCR areas, we can observe the mutual effect of TCR (first peak) and GGR (second peak). (C) GGR's contribution increases whilst the impact of TCR becomes less important towards the end of the gene. (D) Non-TCR regions are solely repaired by GGR. Therefore, repair is expected at later time points during the process.

genomic contexts. Our outcomes are consistent with known influencing factors such as transcription rate and nucleosome density. Remarkably, the clearest link was established between the repair dynamics within genes and their length. To our knowledge, this is an unreported finding for budding yeast. In the following sections, we discuss the relevance of our approach and results within the context of previous publications.

## Applying the CPD repair model

Several studies proposed temporal models for UV-induced lesion repair on different levels of detail. [30] represented NER kinetics in human cells using a Markov-Chain Monte-Carlo approach. It explains the removal of 6–4 photoproducts on a single-cell scale through the random and reversible assembly of repair complexes. A similar model was proposed by [31]. Interestingly, though, they derive very different conclusions, as they suggested that random or pre-assembly of repair proteins is unfavourable. Despite a great level of detail of both models, they are incapable to make region specific predictions. Moreover, as both models are based on microscopy data, they do not explain temporal changes in genome-wide sequencing data on a

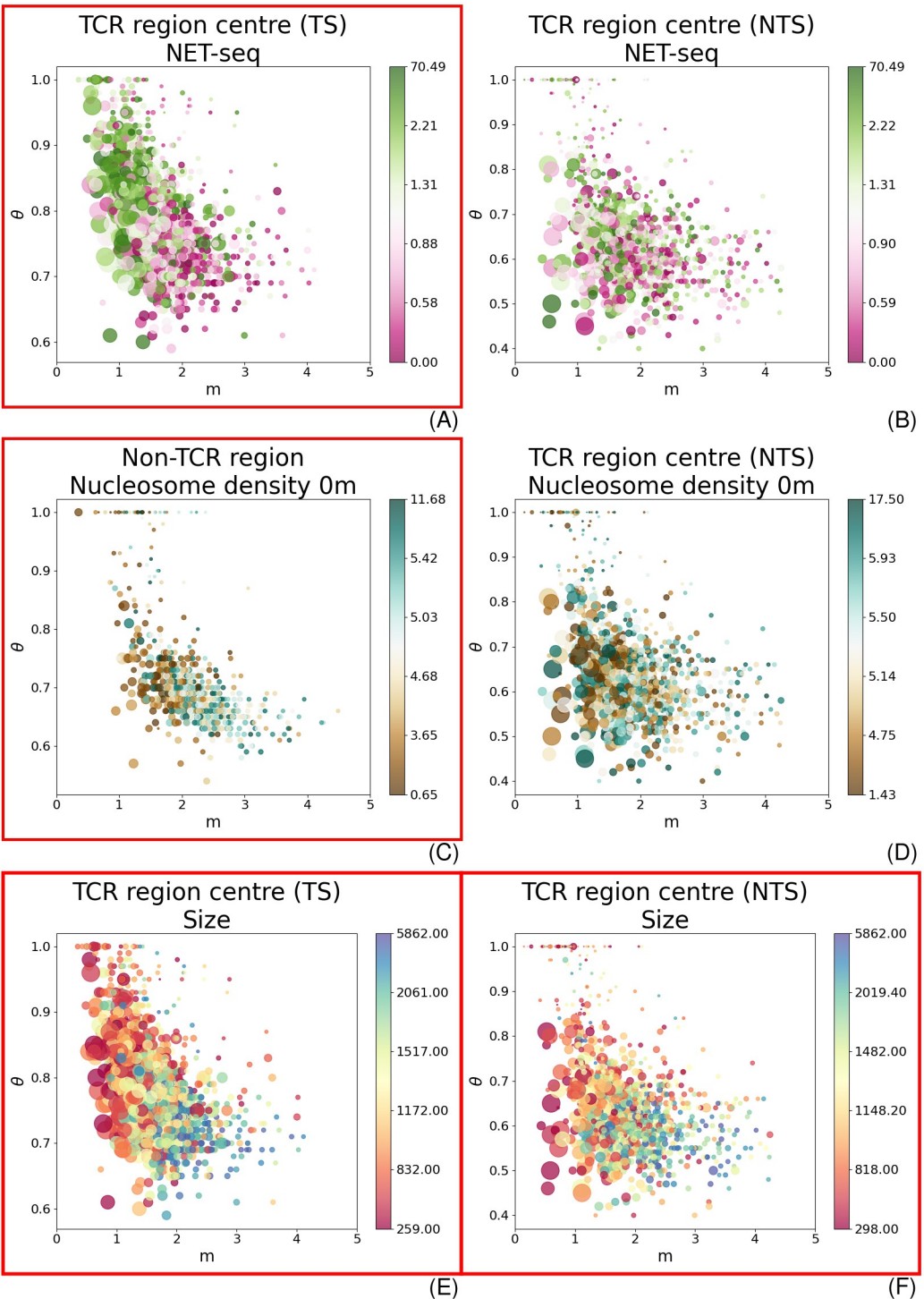

**Fig 4. The parameter distribution is coloured with respect to different genomic properties.** The x and y-axis give the values of $m$ and $\theta$, respectively. The size of the circles show $1/\tau$: the larger the circle, the shorter the characteristic time. Significant interrelationships are marked with a red frame. (A) and (B) are coloured with respect to NET-seq data (pink/low to green/high) for the centre of TS and NTS; (C) and (D) indicate the nucleosome density (turquoise/low to brown/high) for non-TCR regions and the NTS centre; (E) and (F) show the distribution with respect to the TU length (red/small to blue/big) for the centre of TS and NTS.

population scale. A Monte Carlo approach to explain the damage distribution and subsequent repair induced by ionised irradiation in a single cell was proposed by [32]. It also incorporates the collective effect of NER and base excision repair (BER), therefore accounting for potentially competing mechanisms. It should be stressed that the lesion type is considerably different. Again, the predictions are location unspecific. [33] provided a different angle by presenting a protein-protein interaction landscape of NER components in yeast. Predictions about the repair efficiency in different regions were not established. To our knowledge, our model is the first that accounts for region specific changes in population-based data.

As there are only three time points, a data-driven machine learning model is prone to overfitting. In order to find a reasonable representation of the data, we incorporate explicit suppositions to derive Eq 3. To be precise, we presume that repair times follow a Poisson point process, and the non-constant rate $\lambda(t)$ can be described by a mixture of protein diffusion and repair times. This restricts the trajectory to an S-shape. Our understanding of the process makes two independence assumptions. Firstly, there is non-interfering DNA repair between cells; and secondly, protein diffusion is independent of lesion accessibility or repair time.

It is important that the points in a Poisson process are (sufficiently) independent from each other [34]. The repair times that are distributed along the timeline (e.g. as in Fig 1B) should hence symbolise values of independent stochastic variables. We find the presumption of non-interfering repair between cells trivial. *Saccharomyces cerivisiae* are single-cell organisms and should thus react independently to DNA damage. Moreover, yeast cultures were grown in rich medium after UV treatment, precluding any limitations for growth [8]. We conjecture the independence presumption to be reasonable.

The Poisson process is governed by the rate parameter $\lambda(t)$. It is presumed that nearby genomic positions posses similar rates. Therefore, $\lambda(t)$ should have a slow spatial variation, and a segmentation of the CPD-seq data into similar behaving areas is possible. A similar binning approach was applied by [8]. The simplest model sets $\lambda$ to a constant and therefore independent of time. In such a setup, we would expect to observe the largest signal change right after irradiation which subsequently slows down (S13A Fig). This could indeed resemble the beginning of the TS of TCR regions (Fig 3A). However, the majority of areas exhibits the strongest repair rates between 20 and 60 minutes. We conclude that a non-constant $\lambda(t)$ provides a broader applicability for explaining CPD repair.

It remains to find a description of the function $\lambda(t)$. Repair happens through the collective working of several proteins, which need to move through the environment and find their target. Nuclear diffusion has been studied and modelled in detail in different contexts such as chromatin [15–17] and protein movements [13, 35], including the repair protein Rad4 [19]. It is clear that more DNA-protein interactions are possible during longer time windows, since proteins have more time to travel longer distances to reach their target. The distance is denoted by the random variable $R$ which has the expected squared displacement $< R^2 >= D_{\bar{m}} t^{\bar{m}}$ with diffusion constant $D_{\bar{m}}$ [14, 16, 17, 19, 35]. Consequently, the searched volume is $< R^3 >= k'(D_{\bar{m}} t^{\bar{m}})^{\frac{3}{2}} = D_{\hat{m}} t^{\hat{m}}$, where $k'$ is a scaling constant. We couple the Brownian motion through space with an independent random variable $X$ which symbolises repair time or accessibility to the lesion. We define $< X >= \hat{b}$. As $R$ and $X$ are independent, we can write $< XR^3 >=< X >< R^3 >= bt^{\hat{m}}$, with $b = \hat{b}D_{\hat{m}}$. Substituting in Eq 1 results in Eq 3. We find it remarkable that $\beta$ (representing a mixture of diffusion constant and expected waiting time) tends to be inversely proportional to $m$ which is linked to the abnormality coefficient. Despite quick diffusion ($m$ being large), we observe slow repair rates ($\beta$ being low). We interpret this phenomenon as the repair time/accessibility $X$ dominating the process, making diffusion negligible. This argument can be equally applied when diffusion is seemingly very slow.

The simplest representation of an irreversible transition between two states is given by an S-shaped function which contains at most one inflection point. It is important to emphasise that this function is not necessarily inversely symmetric around this point, meaning that the left side can be differently shaped than the right side. These requirements are fulfilled by the KJMA model, which is therefore a sensible choice (see S7 Appendix and S13 Fig).

### Analysing genomic properties which influence repair kinetics

TCR has been identified as a rapid repair pathway on the TS. Intergenic regions and the NTS exhibit significantly slower lesion removal, which was demonstrated on the genomic scale in yeast and human cells [8, 9, 27, 36, 37]. It remains an unsolved quest to find an interrelationship between TCR efficiency and transcription rate. Whilst the two parameters are indeed assumed to be correlated [8, 9], some studies point out that TCR repairs CPDs efficiently at nearly all genes including those with a low transcription [27]. An in-depth analysis is still missing, and there is no clear consensus on how transcription rate is affecting repair. In this work, we compared the model predictions to gene expression. Our analysis clearly shows a significant correlation on the TS and is therefore supporting the common assumption (S7 Fig).

As repair proteins need to recognise and repair lesions on the DNA, it is conjectured that chromatin organisation can significantly modulate the efficiency of CPD repair [8, 12, 27]. However, previous studies were mostly scrutinising the positioning of damage at nucleosomes. CPD removal was shown to be less efficient at the dyad of strongly positioned nucleosomes in yeast [8]. Moreover, GGR on the NTS was asymmetrically inhibited in yeast and human cells with respect to the position within the nucleosome [27]. Even though nucleosome occupancy after UV treatment was already previously probed, the potential relationship of these data with CPD repair was not directly addressed [12]. Our results demonstrate a significant correlation between repair and nucleosome density in non-TCR regions (Fig 4C). We also discovered a clear influence on the beginning of TCR areas (S8 Fig).

Unexpectedly, our outcomes show a strong correlation between TU length and repair. Differences in transcription shutdown and restart after UV treatment relative to gene size were previously reported for human cells [38]. Both transcription regulation and efficient repair are necessary to orchestrate an effective cellular response to UV light. The restart of transcription to pre-irradiation levels is an important step at the final stages. However, a direct evaluation of lesion removal with respect to gene size was not performed. To our knowledge, this is a new finding for CPD repair in yeast. Due to our data pre-processing (Eq 7), we can rule out that the result derives only from the fact that larger areas have a greater potential to include more damage. This is true due to two reasons. Firstly, we normalised the CPD value in each bin (e.g. beginning of the TS) by the number of pyrimidine dimers in the sequence as described in [8]. Secondly, and more importantly, we want to point out that the quotient in Eq 7 lets any length dependence and normalisation of the binned data vanish. Therefore, the values become automatically comparable due to the design of Eq 7. It should be mentioned though that the regions of interest can become rather small when segmenting the gene into subareas. Influence or noise from neighbouring areas cannot be excluded. However, due to the fact that the same result can be obtained with a different segmentation (S3 Appendix and S9 Fig), we presume that it represents a genuine feature of the CPD removal mechanisms in yeast cells.

Lastly, we investigated a potential link to the distance relative to the centromere and telomere depending on which was closer to the region of interest (S12 Fig). A link to repair has not been proposed to our knowledge, which made it an interesting property to produce verifiable model predictions.

In conclusion, our work opens interesting perspectives for future research on DNA repair mechanisms and influencing genomic factors. New experimental data with increased temporal resolution will help to refine the model and analysis. The approach can be similarly used for other organisms including human cells. Moreover, it can be readily applied to sequencing data of any nuclear process that can be represented as a two-state system, and it is not restricted to repair.

### Introducing the repair space—An alternative understanding of the data

We have discussed the model in detail with respect to a stochastic point process. We want to provide an additional interpretation that is motivated by the physical implications of the KJMA model. Next to assuming independence between cells, we conjecture in the following also independent repair within each cell (S8 Appendix). Moreover, we presume that CPD data were converted by Eq 7.

Considering the two-dimensional grid in Fig 1C, the independence assumptions above permit us to re-order repair positions to patterns. Nevertheless, we restrict the re-grouping to stay within areas of interest which are assumed to behave homogeneously. The growth of patterns in the virtual repair space reminds strongly of the phase transition in solids which is described by the KJMA model.

The creation of these repair patterns can be described by the nucleation rate $n$ (which we link conceptually to the expected waiting/repair time) and a growth speed $G$ (which is in this analogy linked to the diffusion process). In the following, we assume $G$ to be constant in all directions. The transformed volume within $\Delta t$ starting from a single nucleation site is therefore

$$v(\Delta t) = \sigma (G\Delta t)^{m-1}, \tag{4}$$

where $\sigma$ denotes a parameter that describes the shape of the expanding pattern (which would become part of $\beta$). Interestingly, the parameters obtain a slightly different meaning from this point of view. $m$ is the Avrami exponent and characterises the geometry of the area covered by repaired positions after their aggregation. For example, if $m - 1 = 2$, the area corresponds to regular disks in a two-dimensional space. Irregular forms can be expressed with non-integer values [39]. Nevertheless, a direct comparison of a physical shape with a virtual pattern might be difficult to imagine. We therefore advocate another interpretation. $m$ can be understood to express time dependence of the repair process (compare with S14 Fig). A similar notion has been also proposed in the physical context [40]. We believe that such an understanding could possibly permit the inclusion of independent results from the realm of physics.

## Methods and materials

### Parameter estimation and derivative

Eq 3 explains CPD repair as an S-shaped transformation over time. It should be noted that the process has a defined starting point at $t = 0$. By applying the natural logarithm on both sides twice, we obtain

$$\ln \frac{1}{1 - f(t)/\theta} = m \ln t + m \ln 1/\tau. \tag{5}$$

Note that the expression is now continuous over $ln\ t$. Given the data points for repair and by assuming a value for $\theta$, the parameters $m$ and $1/\tau$ can be found by solving the linear regression problem defined in Eq 5 (compare with bottom plots in S14 Fig). $\theta$ was determined

through a systematic parameter search. We started with 0.5 as minimal value for transcribed/ TCR regions and 0.4 for all other. We increased it thereafter by $\Delta\theta = 0.01$, until we reached $\theta = 1.0$. We chose the $\theta$-value that can best describe the repair data together with the corresponding parameters $m$ and $1/\tau$. This is determined by maximising the adjusted $R^2$. It represents the variance in the data that can be explained through the model and can be interpreted as goodness of fit. The derivative of Eq 3 is given by

$$df(t) = \frac{m\theta t^{m-1}}{\tau^m}\left(1 - \frac{1}{\theta}f(t)\right)dt. \tag{6}$$

## Data processing

All experimental data that were analysed in this study comes from public databases (see overview in Table 1). CPD-seq data was taken from [8]. It contains two time courses with samples taken at $t_1 \in \{0, 60\}$min and $t_2 \in \{0, 20, 120\}$min, respectively. The location of transcribed areas was taken from [29]. Data signals were partitioned into different segments, i.e. the TS and NTS of TCR regions as well as non-TCR areas. For the latter, we combined both strands to one group. Consequently, the linear regression problem in non-TCR regions was required to find the best representation for both strands.

CPD-seq fragments were normalised by the number of available pyrimidine dimers, as explained in the supplementary material of [27]. The damage distribution was subsequently transformed into repair in area $a$ through

$$R_a(t) = \frac{\sum_i^N CPD_{a_i}(0) - \sum_i^N CPD_{a_i}(t)}{\sum_i^N CPD_{a_i}(0)}, \tag{7}$$

where $N$ denotes the size of $a$, $CPD_{a_i}(t)$ is the normalised CPD signal at time $t$ and locus $i$ in area $a$, and $t \in \{20, 60, 120\}$min. $a$ is any of the previously described regions (e.g. the start of the NTS). We additionally take it for granted that no new CPD lesions can be induced during repair. Hence, data points were enforced to be greater than or equal to zero and monotonously increasing as a function of time. The rectification is defined by

$$
\begin{aligned}
R_a(20) &= \max\{R_a(20), 0\} \\
R_a(t_i) &= \max\{R_a(t_i), R_a(t_{i-1})\},
\end{aligned} \tag{8}
$$

where $t_i \in \mathbf{t} = (20, 60, 120)$.

All other sequencing data that were used for the correlation analysis were averaged over the size of the area of interest. Start, centre, and end of TS and NTS were linked to the same value to smooth out the potential influence of noise. For example, all subregions of a TS were associated to the same transcription rate. Moreover, both strands were compared to the same data,

**Table 1. Overview over the data sets that were used in this study.**

| Property | Strain | Data type | UV Dose | Reference |
|---|---|---|---|---|
| CPD | BY4741 (WT) | CPD-seq | 125 J/m$^2$ | [8] |
| CPD repair | Y452 (WT w.r.t. repair) | XR-seq | 120 J/m$^2$ | [9] |
| Abf1 | BY4742 (WT) | ChIP-seq | 100 J/m$^2$ (0min) | [12] |
| H2A.Z | BY4742 (WT) | ChIP-seq | 100 J/m$^2$ (0min) | [12] |
| Nucleosome distr. | BY4742 (WT) | MNase-seq | 100 J/m$^2$ (0min) | [12] |
| Transcription rate | YSC001 | NET-seq | - | [28] |

e.g. the TS and the NTS were related to the same nucleosome density. We noticed that the NET-seq signal amplitude decreases as a function of distance from the TSS (S15 Fig). This could possibly induce a TU length-specific bias that is not removed by taking the average over the TU length. We could verify, however, that the NET-seq data strongly correlates with independently probed Pol2 ChIP-seq data [41] (S16 and S17 Figs). We therefore assume that it reasonably represents transcription rate, whilst allowing a direct comparison to the results obtained by [9] (S9 Appendix).

With the exception of nucleosome density, all biological data values possess a biased distribution. They strongly peak around a low value but contain large positive tails. To remove a potential bias introduced by outliers, we limited our analysis to the lower 95th percentile. As this procedure was applied to all data (except nucleosome density), we did not introduce a bias towards a certain model. Rather, we improved comparability. The only exception is the MNase-seq signal, as it is approximately normally distributed. We consider that trimming could introduce a bias rather than removing one.

The relative distance to centromeres ($c'$) or telomeres ($t'$) was measured as follows. Denoting the gene position by $x$, we can define

$$d_{mere} = \frac{2 \min\{|x - c'|, |x - t'|\}}{|c' - t'|}.$$

(9)

We divide only by half of the length since the maximal distance ($d_{mere} = 1$) to both centromere and telomere should be the middle between them two.

## Correlating repair dynamics to genomic contexts

Areas with parameter values outside a reasonable range were excluded from the subsequent analysis. We restricted $m \in [0.5, 5]$ and $\tau \in [20, 200]$. $\theta$ was constrained through the parameter search. Motivation and consequences are discussed in S2 Appendix. The repair parameters were investigated in context of other biological data. We opted for a nonparametric classification $k$-Nearest Neighbour ($k$NN) approach. We grouped biological data into high (class $c = 1$) and low values ($c = 0$), such that both classes contained the same number of samples to remove the difficulty posed by the biased data distribution. To train the machine learning model, the input values $\mathbf{x} = (m, \beta, \theta)$ where normalised so that every dimension was normally distributed with zero-mean and a standard deviation of one. We compared the results for several $k$NN models with $k \in \{5, 10, 20, 50, 100\}$ to remove any model specific bias. A trained model compares an unknown input $\hat{\mathbf{x}}$ to the $k$ closest values of a known data set $\{\mathbf{X}, \mathbf{c}\}$ to predict class $\hat{c}$. We opted for the Euclidean distance as similarity measurement. Here, the $i$-th row of $\mathbf{X}$ is $\mathbf{x_i} = (m_i, \beta_i, \theta_i)$, and $c_i$ is the associated class in $\mathbf{c}$. $\hat{c}$ is determined by a majority vote. For example, if more than 50% of the $k$ neighbouring values are classified as group $c = 1$, then $\hat{c}$ is predicted to be group 1 as well. $k$NN is categorised as nonparametric model which permits the comparison of different results. The performance was measured through calculating the prediction error

$$E = \frac{\#\text{Incorrectly classified samples}}{\#\text{ All samples}}.$$

(10)

This was compared to a random baseline model, for which classes were randomly shuffled to a given parameter triple $(m, \beta, \theta)$ during training. Data $\{\mathbf{X}, \mathbf{c}\}$ were arbitrarily partitioned into learning and testing data sets. Every experiment was independently repeated 100 times to reduce the effect of any potential bias. We consider an interrelationship to be important if the prediction error of the true function is significantly lower than the error of the random model ($p < 0.00001\%$ of a one-sided t-test). Moreover, we require that 90% of the prediction errors

are below $E < 0.5$, which is the expected outcome of an unbiased coin-flipping experiment. This significance must be found in three out of the five evaluated $k$ to indicate an interrelationship.

## Supporting information

**S1 Appendix. Determining TCR Regions.**
(PDF)

**S2 Appendix. Discussing the Effect of Data Transformation and Selection.**
(PDF)

**S3 Appendix. Dividing the Data into Genes versus Intergenic Regions.**
(PDF)

**S4 Appendix. Explaining the Correlation Between XR-seq Data and Repair Rate.**
(PDF)

**S5 Appendix. Discussing the $k$NN Approach.**
(PDF)

**S6 Appendix. Analysing Repair Kinetics in Context of Abf1 and H2A.Z Distribution.**
(PDF)

**S7 Appendix. Comparing the KJMA Model With Other Approximations.**
(PDF)

**S8 Appendix. Discussing the Model in Context of the Physical KJMA Model.**
(PDF)

**S9 Appendix. Investigating a Link Between Transcription Rate and TU length.**
(PDF)

**S1 Table. The number of models per region, before and after applying the requirements for parameter ranges.** IGR abbreviates intergenic regions.
(PDF)

**S2 Table. The DC between XR-seq and repair predictions / data for different experimental configurations.**
(PDF)

**S3 Table. Number of non-random interrelationships between model parameters and sequencing data over $k$.** The table gives the number of $k$NN models that could find a correlation between model parameters and genomic context. $k \in \{5, 10, 20, 50, 100\}$. We defined a link to be significant if at least three out of five $k$ find a non-random interrelationship.—means that data was not used in the given configuration. NET denotes NET-seq data, ND is nucleosome density, and meres give the relative distance to centromeres or telomeres. Suffixes S, C, and E denote start, centre and end of an area. NTCR are non-TCR areas. IGR are intergenic/ non-transcribed regions.
(PDF)

**S1 Fig. Scheme of the segmentation setup.** The circles represent the number of cells with ongoing repair in the region. The arrows indicate the region and direction of transcription. The results in the paper follow the *TCR* setup. Here, only the first gene is considered as TCR area which shows more efficient repair than intergenic regions within the first 20 minutes after UV irradiation. All other parts are labelled as non-TCR region. Therefore, it spans from the

end of the first gene to the end of the second. The *gene* configuration (S3 Appendix) partitions the genome into the traditional notion of transcribed and intergenic regions. The TU positions were determined by [29].
(TIF)

**S2 Fig. Relative repair distribution over genomic areas.** Relative repair in non-transcribed regions is chiefly lower than 20% within the first 20 minutes (88.95%). Genic areas with stronger repair dynamics are thus likely supported by TCR. For all other transcripts, we cannot exclude the possibility that they are exclusively repaired by GGR.
(TIF)

**S3 Fig. Example for model predictions.** Data points are given by solid dots. The blue dashed line represents the repair fraction predicted by the model (left axis). The orange dashed line shows the derivative (right axis). (A) The *SNF6* gene can be well approximated. (B) However, *GEM1* exhibits no repair within the first 20 minutes, which results in a switch-like behaviour. (C) This is better understood when showing the data points after transformation according to Eq 5. A linear regression is difficult since they do not align.
(TIF)

**S4 Fig. Example for model prediction and XR-seq data over time.** XR-seq data (points) and the predicted repair rate (dashed lines) are exemplified for genes *BDH1* (orange) and *BDH2* (blue). When re-scaling XR data and repair rate prediction between 0 and 1, both follow clearly similar trends. *BDH2* has its largest XR-seq value at 20 minutes post-irradiation, whereas *BDH1* shows biggest repair rates after 5 minutes. This is indeed captured by the model.
(TIF)

**S5 Fig. The evolution of repair along the NTS.** The average repair evolution for the NTS (blue dashed line) shows a much lower repair fraction ($\approx$0.6) than the other areas. Moreover, the repair rate (orange dashed line) indicates repair at early time points. This could be caused by possible overlapping transcripts or by antisense-transcription-coupled repair. Shaded areas show the standard deviation. The repair trajectory is the same for (A) the beginning, (B) the centre, and (C) the end of the NTS.
(TIF)

**S6 Fig. Example of the learnt function between model parameters and genomic context.** (A) The learnt parameter distribution and the associated class for the true model after applying a principle component transformation. The x and y-axis give the first and second principle component, respectively. Red represent large genes, whereas blue shows low values. (B) The error distribution for the predictions follows the expected outline given by the learnt function in (A). The blue and red circles give values that were classified as short but were actually large and vice versa, respectively. White points are correctly classified. The right bar shows the error distribution along the colour axis, i.e. over estimated, correctly classified, and underestimated values from top to bottom. The lower histogram shows the distribution of overall correctly and incorrectly classified values. (C, D) The learnt parameter map and the error distribution of the random model.
(TIF)

**S7 Fig. Model parameters with respect to transcription rate.** Our results for the transcription rate support the hypothesis that it influences repair on the TS. The x and y-axis give the values of $m$ and $\theta$, respectively. The size of the circles show $1/\tau$: the larger the circle, the shorter the characteristic time. Significant interrelationships are marked with a red frame.
(TIF)

**S8 Fig. Model parameters with respect to nucleosome density.** Nucleosome density is seemingly influencing repair in non-transcribed/non-TCR regions as well as the beginning of the TCR TS and the TS in the *gene* setup. The x and y-axis give the values of $m$ and $\theta$, respectively. The size of the circles show $1/\tau$: the larger the circle, the shorter the characteristic time. Significant interrelationships are marked with a red frame.
(TIF)

**S9 Fig. Model parameters with respect to size.** The size is clearly influencing repair for both, TS and NTS in the *TCR* and *gene* configuration. The x and y-axis give the values of $m$ and $\theta$, respectively. The size of the circles show $1/\tau$: the larger the circle, the shorter the characteristic time. Significant interrelationships are marked with a red frame.
(TIF)

**S10 Fig. Model parameters with respect to Abf1.** The results for Abf1 are more ambiguous. Although we can find a significant correlation to non-TCR regions as expected in the *TCR* setup, the picture is less clear for the *gene* configuration. The x and y-axis give the values of $m$ and $\theta$, respectively. The size of the circles show $1/\tau$: the larger the circle, the shorter the characteristic time. Significant interrelationships are marked with a red frame.
(TIF)

**S11 Fig. Model parameters with respect to H2A.Z.** Similar to Abf1, the correlations with H2A.Z do not allow a straightforward interpretation. Whilst repair in all areas in the *TCR* configuration is seemingly linked to H2A.Z, this tends to be restricted to the TS and NTS in the *gene* setup. The x and y-axis give the values of $m$ and $\theta$, respectively. The size of the circles show $1/\tau$: the larger the circle, the shorter the characteristic time. Significant interrelationships are marked with a red frame.
(TIF)

**S12 Fig. Model parameters with respect to cetromeres and telomeres.** With the exception of the TS in the *gene* setup, the distance to telomeres or centromeres (shortened with *meres*) does not affect repair dynamics. The x and y-axis give the values of $m$ and $\theta$, respectively. The size of the circles show $1/\tau$: the larger the circle, the shorter the characteristic time. Significant interrelationships are marked with a red frame.
(TIF)

**S13 Fig. Comparison of the KJMA model with other functional descriptions.** (A) A homogeneous Poisson repair process with $\lambda(t) = c$ has the strongest change in the beginning which subsequently flattens out. In most investigated regions, such a behaviour is not observed. (B) We compared the performance of different models to describe the data, which is exemplified for gene *LDB16* (*YCL005W*). Black dots represent the repair data (converted CPD-seq data, see Eq 7), whereas the best fit of each model is given in dashed lines. (C) We applied the mean-squared error (MSE, S7 Appendix Eq 1) to compare the performance of the models with respect to the data. The KJMA model and the Hill equation perform undoubtedly better than simpler models like linear or logistic regression. Nevertheless, the Hill equation describes the data slightly yet significantly better. It should be emphasised though that the performance difference is marginal. Width of the shaded areas represents the number of genes that yielded the corresponding error, which is mirrored at the vertical line. The centre horizontal line with the corresponding numbers give the error median. The top and bottom horizontal lines show maximum and minimum, respectively. (D) Despite the fact that non-TCR regions are not expected to show an observable impact by TCR, the Hill equation indicates two mechanisms that act at different time points. Dashed lines give the mean whereas the shaded areas show the

standard deviation. Together with the fact that there is no straightforward interpretation of the Hill equation in context of repair evolution, we conclude that Eq 3 is a sensible choice.
(TIF)

**S14 Fig. Example of the KJMA model.** The KJMA model includes two governing parameters (as the original model does not involve $\theta$), which are exemplified in (A) for $m$ and (B) for $\tau$. Eq 3 can be conviniently converted to a linear regression problem which is shown for the parameter settings of (A) in (C) and for the parameters of (B) in (D).
(TIF)

**S15 Fig. Example of sequencing data representing transcription rate.** The example of the NET-seq signal in comparison to the Pol2 ChIP-seq data probed by [41] shows that Pol2 exhibits a constant augmentation of the signal amplitude at transcribed regions, whereas NET-seq data decrease as a function of distance from the TSS. The Pol2 data is coloured in green, whereas NET-seq is given in blue (light blue represents the Watson, and dark blue is the Crick strand). The example is given for chromosome II around *CHS2* and *CHS3*.
(TIF)

**S16 Fig. Correlation between size and transcription rate.** The plots show the two-dimensional histogram distribution of TU length and different measurements of transcription rate. The number of genes per bin is given through the colour intensities and white numbers. The DC (S4 Appendix Eq 3) per measurement is given in the title. Size and transcription rate data was divided into 50 equally sized bins, which is given by the x and y axis. We use the 95th percentile to remove strong outliers. (A) The histogram distribution of NET-seq transcription with respect to size reveals that smaller genes tend to have higher transcription rates than larger genes. (B) This link is weakened when considering Pol2 ChIP-seq data.
(TIF)

**S17 Fig. Correlation between NET-seq and Pol2 ChIP-seq data.** The plot shows the two-dimensional histogram distribution of NET-seq and Pol2 data. The number of genes per bin is given through the colour intensities and white numbers. Size and transcription rate data was divided into 50 equally sized bins, which is given by the x and y axis. NET-seq data and Pol2 ChIP-seq signal are strongly related (DC = 0.75, S4 Appendix Eq 3). As we consider only two groups of genes with respect to transcription, i.e. genes with a low (blue) or high transcription rate (red), we can confirm that the majority of regions fall into the same category.
(TIF)

**S18 Fig. Model predictions with respect to XR-seq data in the *gene* setup.** (A) The model predictions in the *gene* configuration are less correlated with the XR-seq data than in the *TCR* setup (DC = 0.241, S4 Appendix Eq 3). (B) When correlating the relative repair rate and the XR-seq data in the *gene* setup, the DC is as low as for the model predictions (DC = 0.231). Therefore, we assume that the weak linkage is due to the data segmentation.
(TIF)

## Acknowledgments

We thank Zoë Slattery for proofreading the manuscript.

## Author Contributions

**Conceptualization:** Leo Zeitler, Cyril Denby Wilkes, Arach Goldar, Julie Soutourina.

**Data curation:** Leo Zeitler, Cyril Denby Wilkes.

**Formal analysis:** Leo Zeitler, Arach Goldar.

**Funding acquisition:** Julie Soutourina.

**Investigation:** Leo Zeitler, Cyril Denby Wilkes, Arach Goldar, Julie Soutourina.

**Methodology:** Leo Zeitler, Arach Goldar.

**Project administration:** Julie Soutourina.

**Resources:** Cyril Denby Wilkes, Arach Goldar, Julie Soutourina.

**Software:** Leo Zeitler.

**Supervision:** Cyril Denby Wilkes, Julie Soutourina.

**Validation:** Leo Zeitler, Cyril Denby Wilkes, Arach Goldar, Julie Soutourina.

**Visualization:** Leo Zeitler.

**Writing – original draft:** Leo Zeitler, Cyril Denby Wilkes, Arach Goldar, Julie Soutourina.

**Writing – review & editing:** Leo Zeitler, Cyril Denby Wilkes, Arach Goldar, Julie Soutourina.

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
