## [Decision Letter · Decision Letter 0]

25 May 2022

Dear Soutourina,

Thank you very much for submitting your manuscript "A Quantitative Modelling Approach for DNA Repair on a Population Scale" for consideration at PLOS Computational Biology.

As with all papers reviewed by the journal, your manuscript was reviewed by members of the editorial board and by several independent reviewers. In light of the reviews (below this email), we would like to invite the resubmission of a significantly-revised version that takes into account the reviewers' comments.

As you will see, both reviewers have considerable criticism regarding the application of the KJMA model to DNA repair. While the first reviewer believes that this application is flawed, the second reviewer also has significant concerns. Hence, why the assumptions of the KJMA model are fullfilled in this case, and why it can be applied to this biological question should be justufied more thoroughly. Reviewer 2 also recommends putting less emphasis on the KJMA model and checking if an alternative model yielding a S-shaped response might also be considered. Hence we urge you to seriously consider these concerns and see if and how you can address them. Please also consider the other comments, for example regarding the small number of timepoints in your application.

We cannot make any decision about publication until we have seen the revised manuscript and your response to the reviewers' comments. Your revised manuscript is also likely to be sent to reviewers for further evaluation.

Sincerely,

Carl Herrmann, Ph.D.

Associate Editor

PLOS Computational Biology

Ilya Ioshikhes

Deputy Editor

PLOS Computational Biology

Reviewer's Responses to Questions

**Comments to the Authors:**

Reviewer #1: I wonder if there is a flaw in the analogy between the physical role of the KJMA model and how it is being superimposed on the biological pattern of repaired sites. I find the argument that underlies figure 1 to be problematic: the assumption that the repair is independent between cells and between sites within cells, which is required to gather the repaired sites into the contiguous block 'pattern', needs much more discussion and evidence. But even then, it is apparent that the spatial metric properties of the equation in the physical application of this equation means that in the domain for which it was derived, the KJMA equation makes a strong use of the fact that one can't freely permute the spatial coordinates arbitrarily. The paper as it stands doesn't really resolve this conflict - it's possible that a major re-phrasing of the paper so that the KJMA model is entirely motivated by the biological setting would resolve this (it's unnecessary to mention the physicists' use of the equation in anything but the most fleeting way, if there is an inherently biological motivation for it), but the situation where phase regions can only grow on their boundary, for instance, doesn't have a clear translation to me when phrasing in terms of repair sites: after the 'pattern' is formed as in figure 1, why are some sites 'special' (appearing on the boundary) whilst others aren't.

It is entirely possible that the KJMA model merely describes a stochastic point-process of repair, which seems a much simpler and more intuitive explanation, and that the time-profile of fraction of repaired sites is purely an emergent phenomenon: early on the proportion of repaired is small because of the lack of accumulated time in which to undergo repair relative to the number of damaged sites, and that as time progresses the accumulation increases but then starts to tail off as the number of available unrepaired damage sites becomes sparser. There is nothing inherently two-dimensional about this point-process approach, and if it arrives at a similar or identical functional form for the time dependence of the repair rate, then the rather forced analogy could be removed.

The use of KNN in the correlation of parameter estimates to biological features seems overly and unncessarily complex - a data-dependent binary split of the data (it's not clear if this is done on the parameter estimates or the biological features) and then doing a t-test seems prone to measurement error bias - a non-parametric test of association between the two continuous variables would seem more straightforward, or a spline regression if a non-linear association was suspected.

I found the impact of the concerns about the applicability of the KJMA model overshadowed my reading of the rest of the paper. The testing of association between biological features and repair rates would have been more convincing if it had been done directly (perhaps empirically, even a seriation or simple PCA decomposition of the signal clustering being graphically annotated with the biological feature) in addition to going via the correlation with KJMA model parameters.

Thank you for supplying the github code - I would recommend a DOI service so that a more permanent record can be associated with the code.

Reviewer #2: The authors consider a quantitative treatment of the dynamics of DNA repair via an application of the KJMA model. Parameters of the model are tested for correlation with various biological quantities such as transcription, finding a novel connection between repair dynamics and gene size.

Some aspects of the approach and results are, to the best of my knowledge, novel. Moreover, I believe that the findings presented will be of interest to the field. However, I do have some concerns regarding interpretation and presentation of the work.

Major Concerns:

1. Connection to the KJMA model

I actually find the mapping of DNA repair to the KJMA model a bit of a stretch. The authors assume (reasonably) independence of repair within and between cells. They also claim that this allows DNA repair to be modelled using the KJMA model as new regions of repair can be rearranged to give a growing pattern on a two-dimensional grid. However, it seems that independence really means that DNA repair should be modelled as independent nucleation events that do not grow and much of the connection to KJMA model would seem to be lost. The authors should either strengthen the justification for the use of the KJMA model or remove some of the emphasis on its use from the manuscript. If the assumption is that nucleation is time dependent such that an S-shaped curve is expected for the progression of repair, then, in the absence of any further information, any simple model that produces an S-shaped response should be applicable. It might be reasonable to suggest employing the KJMA model as it is well known and produces an appropriately shaped response curve, or perhaps the authors can show that the KJMA model is capable of treating the case of time-dependent nucleation without growth.

2. Small number of time points

The authors do acknowledge that an important future step is to apply the modelling approach to data with greater time resolution. Here, they apply a three-parameter model to a time course with only four points. Arguably, they have a sizeable data set in that the data is genome wide, covering many regions of repair which can be partitioned into distinct types. However, this is quite a low number of points for each individual fit. I would recommend that the authors provide some additional analysis on model selection, ideally showing that the S-shape curve provides a significantly better fit to the data than the simplest alternative (which would be a linear model). Alternatively, the correlation results could be compared to a similar approach with a linear model, ideally showing that the approach with an S-shape curve is more powerful in revealing links between the chosen biological quantities and the dynamics of repair.

3. Potential problem with the fitting algorithm

The authors claim that for data showing no (or negative) repair levels at 20 minutes obtaining good fits is difficult (Figure S2B and associated discussion). I can estimate what seems to be a superior fit to YAL048C with values of theta = 0.7, m = 6, tau = 45 (all within the allowed ranges). The authors should provide more reasoning as to why the apparently poor fit in Figure S2B was chosen by the algorithm.

4. Overall presentation

The manuscript was quite difficult to follow in places, often putting heavy emphasis on minor details and giving limited room to what I considered to be important points. I recommend some streamlining of the manuscript to improve readability.

Minor Concerns:

5. The authors state that TCR is not expected to function on the non-transcribed strand. Has any potential repair coupled to non-coding antisense transcription been considered here?

6. Figure 2 – the top and bottom panels appear to have been switched compared with what is described in the main text and the figure caption.

7. Some of the description in the main text is more suited for the methods section and some of the description in the methods section might be more suited for the main text.

8. Figure 6 is missing a caption title (all other main-text figures have this).

9. Figure 4 – the legend spots were too small for me to easily distinguish between the colours.

10. Figure 3 – the colour scheme is confusing: I don’t think blue and orange should be reused for the lines at the bottom as they appear to represent something different there.

11. Figure 3 – I think it would be helpful to also describe additional classifications in this schematic, including TS, NTS, beginning, centre and end.

12. Figure 5 – the meaning of the lines and shaded areas should be in the figure caption. How the averages were taken should also be described in sufficient detail somewhere in the manuscript.

13. Figure 5 (and connected discussion) – Is there some analysis that could be done in order to separate out early- and late- repair subpopulations (assuming that they are indeed what is driving the double-peak behaviour in the derivative).

14. References appear not to have been ordered.

15. Line 180 – “transcripts that exhibit strong repair …” – should this be transcribed regions as the transcripts themselves are not repaired?

16. Many of the supplementary figure captions are missing descriptions (for example, the meaning of the size of the circle points in the plots).

**Have the authors made all data and (if applicable) computational code underlying the findings in their manuscript fully available?**

Reviewer #1: Yes

Reviewer #2: Yes

PLOS authors have the option to publish the peer review history of their article (what does this mean?). If published, this will include your full peer review and any attached files.

Reviewer #1: **Yes: **Gavin Kelly

Reviewer #2: No
---

## [Decision Letter · Decision Letter 1]

27 Jul 2022

Dear Soutourina,

Thank you very much for submitting your manuscript "A Quantitative Modelling Approach for DNA Repair on a Population Scale" for consideration at PLOS Computational Biology. As with all papers reviewed by the journal, your manuscript was reviewed by members of the editorial board and by several independent reviewers. The reviewers appreciated the attention to an important topic. Based on the reviews, we are likely to accept this manuscript for publication, providing that you modify the manuscript according to the review recommendations.

Please carefully consider the recommendations of the reviewer in your revised manuscript, especially regarding the labelling of some figures.

Sincerely,

Carl Herrmann, Ph.D.

Associate Editor

PLOS Computational Biology

Ilya Ioshikhes

Deputy Editor

PLOS Computational Biology

[LINK]

Reviewer's Responses to Questions

**Comments to the Authors:**

Reviewer #2: The authors have undertaken a substantial rewrite of many sections of the main text and additional work to address the reviewers’ concerns. I now find the paper to be much improved, with the main message being much clearer. I also now believe I have a firmer grasp on the authors’ original intent with regards to the abstract connection with the KJMA model.

Recasting the direct connection to the biological phenomenon of DNA repair as a Poisson point process as the first reviewer suggested (essentially what I was trying to describe as time-dependent nucleation without spreading, using the terminology of the KJMA model) and then connecting this to the KJMA model makes things much more believable from a biological perspective.

I was pleased to see that the authors had done additional analysis to compare the overall quality of fit between a number of simple candidate models to show that their approach is significantly better than the simplest models. When combined with the clearer explanation of the connection to the biology, this sufficiently justifies the use of the KJMA model in my opinion.

I now understand why the seemingly poor fit to YAL048C was chosen by the algorithm. I still think there might be issues with fitting equally to a data point which has been set to (or close to) zero due to the measurement being negative (an impossibility in the model). Such points could have larger errors on them than positive measurements. Perhaps a fitting algorithm that takes differences in errors into account would improve matters and allow more genes to be considered for further analysis. I appreciate that this would entail significant extra effort and be beyond the scope of this manuscript, but it is something that could potentially be covered in future work.

There are a few minor issues that I think should be addressed before publication.

1. I spotted a few possible typos when reading the revised manuscript:

a. Line 6: ‘evolutionary’ to ‘evolutionarily’?

b. Line 98: ‘lager’ to ‘larger’ ?

c. Line 152: ‘region’ to ‘regions’ ?

2. S3 Fig: Caption seems to be missing completely.

3. S4 Fig/Caption: The examples that are being used should be stated/described.

4. S13 Fig Caption:

a. For panel B, for absolute clarity, that this is a single-gene example should be stated and the black dots need to be described as the experimental data points.

b. For panel C, again for absolute clarity, MSE should be defined and the meaning of the lines/bars and shaded regions in the plots should be described.

5. S16 Fig/Caption:

a. The kind of correlation calculated should be stated/described somewhere

b. For clarity and ease of reading, the meaning of the numbers should be stated/described (also applies to S17 Fig Caption).

c. The units of the axes should be described (also applies to S17 Fig)

**Have the authors made all data and (if applicable) computational code underlying the findings in their manuscript fully available?**

Reviewer #2: Yes

PLOS authors have the option to publish the peer review history of their article (what does this mean?). If published, this will include your full peer review and any attached files.

Reviewer #2: No

Figure Files:

Data Requirements:

Reproducibility:

References:

---

## [Editor Report · Decision Letter 2]

12 Aug 2022

Dear Soutourina,

We are pleased to inform you that your manuscript 'A Quantitative Modelling Approach for DNA Repair on a Population Scale' has been provisionally accepted for publication in PLOS Computational Biology.

Best regards,

Carl Herrmann, Ph.D.

Associate Editor

PLOS Computational Biology

Ilya Ioshikhes

Deputy Editor

PLOS Computational Biology

---

## [Editor Report · Acceptance letter]

5 Sep 2022

PCOMPBIOL-D-22-00499R2 

A Quantitative Modelling Approach for DNA Repair on a Population Scale

Dear Dr Soutourina,

I am pleased to inform you that your manuscript has been formally accepted for publication in PLOS Computational Biology. Your manuscript is now with our production department and you will be notified of the publication date in due course.

With kind regards,

Katalin Szabo
